# Collagen Membrane as Water-Based Gel Electrolyte for Electrochromic Devices

**DOI:** 10.3390/gels9040310

**Published:** 2023-04-06

**Authors:** Carmela Tania Prontera, Nunzia Gallo, Roberto Giannuzzi, Marco Pugliese, Vitantonio Primiceri, Fabrizio Mariano, Antonio Maggiore, Giuseppe Gigli, Alessandro Sannino, Luca Salvatore, Vincenzo Maiorano

**Affiliations:** 1CNR NANOTEC—Institute of Nanotechnology c/o Campus Ecotekne, University of Salento, Via Monteroni, 73100 Lecce, Italy; roberto.giannuzzi@nanotec.cnr.it (R.G.); marco.pugliese@nanotec.cnr.it (M.P.); vitantonio.primiceri@unisalento.it (V.P.); fabrizio.mariano79@gmail.com (F.M.); antonio.maggiore@nanotec.cnr.it (A.M.); giuseppe.gigli@unisalento.it (G.G.); vincenzo.maiorano@nanotec.cnr.it (V.M.); 2Department of Engineering for Innovations, University of Salento, Via Monteroni, 73100 Lecce, Italy; alessandro.sannino@unisalento.it (A.S.); luca.salvatore@unisalento.it (L.S.); 3Department of Mathematics and Physics “Ennio De Giorgi”, University of Salento, Via per Arnesano, 73100 Lecce, Italy

**Keywords:** membrane electrolyte, hydrogel electrolyte, type I collagen, protonic conductivity, electrochromic devices

## Abstract

Bio-based polymers are attracting great interest due to their potential for several applications in place of conventional polymers. In the field of electrochemical devices, the electrolyte is a fundamental element that determines their performance, and polymers represent good candidates for developing solid-state and gel-based electrolytes toward the development of full-solid-state devices. In this context, the fabrication and characterization of uncrosslinked and physically cross-linked collagen membranes are reported to test their potential as a polymeric matrix for the development of a gel electrolyte. The evaluation of the membrane’s stability in water and aqueous electrolyte and the mechanical characterization demonstrated that cross-linked samples showed a good compromise in terms of water absorption capability and resistance. The optical characteristics and the ionic conductivity of the cross-linked membrane, after overnight dipping in sulfuric acid solution, demonstrated the potential of the reported membrane as an electrolyte for electrochromic devices. As proof of concept, an electrochromic device was fabricated by sandwiching the membrane (after sulfuric acid dipping) between a glass/ITO/PEDOT:PSS substrate and a glass/ITO/SnO_2_ substrate. The results in terms of optical modulation and kinetic performance of such a device demonstrated that the reported cross-linked collagen membrane could represent a valid candidate as a water-based gel and bio-based electrolyte for full-solid-state electrochromic devices.

## 1. Introduction

One of the most important components in an electrochemical device is the electrolyte since it is responsible for ionic transportation within the device, thus influencing its performance. Traditional liquid electrolytes are not suitable for the evolution of electrochemical devices since they show several disadvantages, such as the employment of corrosive and inflammable solvents, leakage and evaporation problems, and unfit for flexible devices [1]. For these reasons, there is a growing interest in the development of new electrolytes suitable for full-solid-state devices since they allow at the same time to improve device safety, simplify the fabrication process, and achieve flexible devices [1]. In this context, polymers represent promising candidates [2]. In particular, gel polymer electrolytes contain liquid electrolytes immobilized in a host made up of one or more polymer matrices [2,3,4]. Their features are excellent mechanical integrity, film-forming ability, easy processability, and higher ionic conductivity compared to solid-state polymer electrolytes [2,3,5]. Among the liquid electrolytes that can be immobilized into a polymeric matrix, aqueous solutions represent an interesting alternative for the development of safe and non-toxic gel polymer electrolytes. Hydrogels are three-dimensional (3D) polymeric networks composed of hydrophilic-functional groups allowing them to absorb and retain a large amount of water without dissolving [6]. Therefore, hydrogels can hold a large amount of aqueous ionic solution providing good ionic conductivity, keeping at the same time good mechanical stability originating from the polymeric network [6]. Different synthetic polymers have already been employed to obtain hydrogels, but natural polymers have gained attention as promising substitutes for traditional synthetic polymers thanks to their advantageous properties, such as biodegradability, biocompatibility, non-toxicity, sustainability, etc. [7,8]. Indeed, global warming, price fluctuations, reduced oil resources, and pollution are just some of the factors that are pushing toward increasing use of eco-friendly biomaterials.

Natural materials directly extracted from biomass resources, such as polysaccharides, proteins, and lipids, represent one of the most relevant categories. In particular, polysaccharides are the most studied bio-polymeric electrolytes for their large availability, abundance, and accessibility. For all these reasons, they were adopted in the fabrication of eco-friendly devices [9,10]. Among proteins, soybean protein, gelatin, and collagen were used as matrices for gel polymeric electrolytes aiming for more sustainable solid-state electrochemical devices [11,12,13,14]. In particular, gel polymeric electrolytes were obtained by saturating a membrane of soybean protein with an aqueous solution of Li_2_SO_4,_ and such electrolytes were employed for the fabrication of solid-state electric double-layer capacitors [11]. Similarly, a gelatine-based gel electrolyte consisting of porcine skin-derived gelatine and sodium chloride was employed in screen-printed and stencil-printed supercapacitors [12].

In this context, type I collagen is the most abundant structural protein of vertebrates’ connective tissue, where it provides strength and structural stability, performing regulatory functions [15,16]. Its unique fingerprint consists of three left-handed polyproline-II helices of about 1000 amino acid residues that assemble in a right-handed triple helix [16,17,18]. Each polyproline-II chain is characterized by the replication of the (Gly-X-Y)_n_ triplet, where Gly is glycine, and the X and Y positions are usually occupied by proline and hydroxyproline, respectively [19]. In this recurrence, glycine plays a fundamental role in the three α helices packing, while proline and hydroxyproline are crucial elements in the triple-helical-structure stabilization [20]. Thanks to its advantageous intrinsic properties, such as biodegradability, biocompatibility, bioactivity, easy manufacturing, and customizable properties, collagen is one of the most used biomaterials for healthcare applications [20,21,22,23,24,25]. Collagen harvesting mostly relies on its extraction from animal tissues by-products of the food industry (i.e., skin, tendon, scales, cartilage, bone, and so on) [21,26]. The recovery and valorization of waste materials make collagen eco-friendlier and more cost-effective than other approaches [27]. Recently, aside from applications in the biomedical, pharmaceutical, cosmetic, and food fields, collagen started to be investigated in the field of energy devices as a sustainable source for nanoporous carbon materials exploitable as batteries anode [28]. Regarding the application of collagen as a polymeric network for hydrogel-based electrolytes, a collagen fiber membrane infiltrated with Na_2_SO_4_ aqueous solution showed an ionic conductivity of about 9 × 10^−3^ S/cm and it was used as an electrolyte in an electrical double-layer capacitor [13]. Collagen was also employed as an electrolyte of fuel cells in humidified conditions [14]. Indeed, Matsuo et al. demonstrated that the proton conductivity of the collagen membrane goes from 1 × 10^−5^ to 4 × 10^−3^ S/cm when the relative humidity is increased from 53% to 100%. Such an effect is due to the formation of water bridges bonded with the collagen peptide chains, inducing an increase in proton conductivity compared to a dry-collagen membrane.

Starting from such a background, in this work, uncrosslinked and cross-linked collagen membranes derived from equine tendons have been investigated as alternative gel-like polymeric electrolytes for the fabrication of electrochromic devices. Because of produced uncrosslinked collagen membranes’ low degradation resistance in acid environments, a physical cross-linking method was chosen and applied among cross-linking strategies to enhance collagen substrate properties without reducing their eco-friendly character and their biocompatibility profile. In particular, a heat-mediated cross-linking method was selected to induce the formation of amide and ester bonds via condensation reactions [29]. The swelling capability and the mechanical properties of the two kinds of membranes were evaluated, and the cross-linked collagen proved to be the best candidate for the preparation of the gel electrolyte. The ionic conductivity and the optical properties of the membrane after impregnation with H_2_SO_4_ aqueous solution were measured, demonstrating its suitability for the fabrication of electrochromic devices thanks to its good transparency and excellent conductivity. An electrochromic device was fabricated exploiting such a polymeric gel electrolyte, showing good modulation properties and excellent kinetic properties.

These results demonstrate the potential of eco-friendly and biodegradable collagen-based membranes as new and effective hydrogel-like electrolytes for the fabrication of full-solid-state electrochemical devices and, in particular, electrochromic devices.

## 2. Results and Discussion

Uncrosslinked (UN) and cross-linked (DHT) collagen membranes were prepared by air-drying; their IR spectra are reported in Figure 1. The peaks of amide I, amide II, and amide III of type I collagen were detected, as well as amide A and B contributions [30,31]. The amide I (1621–1635 cm^−1^) band is associated with C=O hydrogen-bonded stretching, the amide II (1535–1548 cm^−1^) peak is associated with C-N stretching and N-H in-plane bending from amide linkages, and the amide III (1220–1240 cm^−1^) to the N-H bending [32,33,34,35,36]. The peaks at 1400 and 1340 cm^−1^ were assigned to the wagging and deformation modes of -CH_3_ and -CH_2_ of the glycine backbone, besides the proline and hydroxyproline sides [32]. The contributions at approximately 1080 cm^−1^ and 1030 cm^−1^ were attributed to the stretching vibration of C-O-C and C-O, respectively [34]. Lastly, contributions at about 3400–3500 cm^−1^ and 3000–3080 cm^−1^ could be observed and attributable to the amide A and amide B, which are ascribed to the N-H stretching coupled with intramolecular H-bond and N-H bend, respectively [37]. The presence of contributions attributable to type I collagen confirmed that the process employed for collagen-membranes production did not significantly affect the material structural conformation. The peaks of amide I, II, II, A, and B were found to be almost the same in uncrosslinked and cross-linked samples, with a slight shift to lower frequencies for cross-linked samples. In particular, the shift to lower frequencies in the amide I and III of cross-linked samples prompted the involvement of the -C=O and -NH groups in new bonding interactions and, thus, the effectiveness of the applied physical cross-linking treatment [31]. In Figure 1B, a representation of the chemical structure after the cross-linking process is shown, and the new bonds are highlighted in red.

To investigate in-depth collagen secondary structure in the matrices after the production process, five contributions were identified from amide I deconvolution (Figure 2). In particular, β-sheet (peak center: 1610–1642 cm^−1^), random coil (peak center: 1642–1650 cm^−1^), α-helix (peak center: 1650–1660 cm^−1^), β-turn (peak center 1660–1680 cm^−1^), and β-antiparallel (peak center 1680–1700 cm^–1^) components were detected (Table 1) [38,39,40]. The increase of the β-sheets component in cross-linked samples could be explained by assuming the formation of bonds among collagen molecules that are laterally associated that spectroscopically mimic β-sheet structures [41,42]. Moreover, the integrity of the collagen triple-helical unit was evaluated by the amide III/A_1450_ ratio [43]. A ratio equal to or higher than unity confirmed its conformational structure preservation after the production process (uncrosslinked amide III/A_1450_ = 2.1; cross-linked amide III/A_1450_ = 1.9).

Additionally, the –OH stretching band (range 4000–3000 cm^−1^) corresponding to the amide A was analyzed and deconvoluted into four components (Figure 3) whose frequencies were related to different O-H bond lengths (Table 2), which in turn were correlated to the hydrogen bond network around the protein. Differences in hydrogen bond distances could provide information about cross-linking due to protein modifications. Thus, following the procedure of the second derivative analysis, four Gaussian components were identified, corresponding to the four classes of water molecules that can be bound to the protein. Each of them has different vibrational energies and a single average H-bond distance (H···OH length) of 0.31, 0.29, 0.2,8, and 0.25 nm, respectively [41,44]. The defined peaks were found at about 3670, 3460, 3250, and 3110 cm^−1^, according to the literature [41,45]. The sub-band peaking at about 3590–3690 cm^−1^ region corresponds to H-bond distances, characteristic of a vapor-like state, attributed to protein non-H-bonded or weakly H-bonded O–H groups [41,46]. As collagen films were in a dehydrated state, about 4.4% of non-H-bonded O–H groups were found in UN films, while no non-H-bonded O–H groups contributions were found in DHT films. The two-component bands peaking at 3480–3490 cm^−1^ and 3240–3250 cm^−1^ attributed to water molecules coordinated by two or three H-bonds corresponding to water molecules that form inter- or intra-molecular bridges. Indeed, they were found to constitute about 18% (for ν2) and 50% (for ν3) of the total water content for both UN and DHT. As suggested by Bridelli et al., the corresponding H-bond distances suggested that they could be attributed to the hydrogen bonding of –C=O groups belonging to glycine (d(C=O···W) = 0.295 nm) and hydroxyproline (Hyp) (d(C=O···W) = 0.284 nm) hydroxyl residues [41]. Lastly, the component at about 3110 cm^−1^, which was attributed to water molecules hydrogen-bonded to polar and charged groups exposed to the macromolecule surface, was found to be higher for DHT (about 33%) than UN (26%), suggesting the presence of a high number of hydrogen bonds in DHT films. Moreover, a variation was noticed in the band-intensities ratio of –CH_2_ and –CH_3_ (uncrosslinked CH_2_/CH_3_ ratio: 1.6; cross-linked CH_2_/CH_3_ ratio: 6.0); the increase in the band intensity assigned to the methylene groups (2925 cm^−1^) compared to the methyl groups one (2950 cm^−1^) suggested that cross-linking reaction occurred in cross-linked collagen matrices [47].

Considering the application of the collagen film as a hydrogel-based electrolyte, their stability in water and aqueous electrolytes and their water/aqueous electrolyte-absorbing capacity was evaluated. Sulfuric acid was selected as an aqueous electrolyte because of its high ionic conductivity, and three different concentrations were tested. Cross-linked and uncrosslinked collagen membranes were soaked overnight in water and sulfuric acid (1%, 2%, 5% *v*/*v*). Uncrosslinked collagen membrane in water showed a very high water-absorption capability, clearly visible from the significant swelling-degree ratio (close to 10) and thickness increase (dry uncrosslinked film thickness: 55 ± 6 µm; uncrosslinked film thickness in H_2_SO_4_: 856 ± 24 µm) (Figure 4A). Non-significant differences in thickness variation related to sulfuric acid concentration were detected (*p* > 0.5). However, the uncrosslinked matrix turned out not to be stable in sulfuric acid solutions since it lost its film-like structures and almost completely melted in 1–5% *v*/*v* sulfuric acid.

Vice versa, a lower water-absorption capability and a reduced thickness increase were observed for cross-linked membranes (dry cross-linked film thickness: 38 ± 5 µm; cross-linked film thickness in H_2_SO_4_: 242 ± 12 µm; swelling-degree ratio close to 3). Again, non-significant differences in thickness variation related to sulfuric acid concentration were detected (*p* > 0.5). Despite the reduced ability to retain H_2_SO_4_, cross-linked films, unlike non-crosslinked ones, turned out to be able to withstand their structure even after soaking in 5% (*v*/*v*) sulfuric acid. The swelling ratio of the different membranes soaked in different solutions is reported in Figure 4B.

Tensile tests were performed to characterize uncrosslinked and cross-linked collagen matrices, in terms of mechanical properties, after overnight incubation in water and 1–5% (*v*/*v*) H_2_SO_4_. The mechanical properties of uncrosslinked films in H_2_SO_4_ were not assessed because of its almost complete dissolution in sulfuric acid that did not allow for handling and clamping it in a tensile-test-machine tool. As expected, the stress-strain curves of matrices were all characterized by a linear elastic region, followed by a non-elastic region and a rupture region (Figure 5) [31,40]. The constitutive bond of the uncrosslinked matrix was found to be statistically different from that of the cross-linked matrices in terms of E and εr (Table 3). In particular, the elastic modulus of the uncrosslinked matrix (1.2 ± 0.3 MPa) proved to be lower than that of the cross-linked matrices (2.7 ± 0.5 MPa), suggesting a matrix stiffening due to the applied physical cross-linking treatment (*p* = 0.01). Moreover, while σ_max_ was found to be almost the same for both uncrosslinked and cross-linked matrices (*p* = 0.2), the εr value was significantly reduced by the cross-linking treatment (UN-H_2_O treatment: εr = 131 ± 7%; DHT-H_2_O treatment: εr = 67 ± 7%; *p* = 0.0003). As regards the presence of the aqueous electrolyte, it was verified to significantly influence the E value as well as the εr value. Indeed, the sulfuric acid increased cross-linked matrices’ E values and reduced their εr value. In particular, the sulfuric acid concentration was proven to be inversely proportional to E values and directly proportional to εr values (*p* < 0.01). In other words, the increase in sulfuric acid concentration was responsible for the cross-linked matrix mechanical properties lost. Representative snapshots of uncrosslinked and cross-linked collagen matrices at ε% = 0% and at their maximum deformation before the break in the presence of water and aqueous electrolytes are reported in Figure 6.

Considering the reported results, cross-linked membrane proved to be a good candidate to obtain a self-standing hydrogel-like electrolyte that can be easily integrated into a multilayer device; for such reasons, it was further investigated. In particular, to integrate the reported hydrogel electrolyte in an electrochromic device, optical properties and ionic conductivity should be evaluated. Therefore, the ionic conductivity of the cross-linked membrane impregnated with water at different H_2_SO_4_ concentrations (1%, 2%, and 5% *v*/*v*) was evaluated through electrochemical impedance spectroscopy. Nyquist plots of electrochemical impedance spectra of the different membranes, measured at 0 V, have been reported in Figure 7A. The electrolyte ohmic resistance is obtained by the intercept at the real impedance axis of high frequency. The ionic conductivity (see equation in experimental details) of M-Water, M-H_2_SO_4_ (1%), M-H_2_SO_4_ (2%), and M-H_2_SO_4_ (5%) hydrogel electrolytes was found to be 0.004, 0.12, 0.16, and 0.15 mS/cm, respectively. All H_2_SO_4_-impregnated membranes showed higher ionic conductivity than water-based hydrogel.

The ionic conductivity of the membrane impregnated with pure water can be attributed to the intrinsic conductivity of the wet collagen membrane. In particular, it has been largely reported that ionic conductivity in collagen is strongly affected by its hydration state [48]. The water molecules that are absorbed by the membrane interact with the N-H, O-H, and C=O groups of the peptide chains through hydrogen bonds. When a small quantity of water is absorbed, the water molecules are isolated from each other, but when the number of absorbed water molecules increases, firstly, intra-helix bridges and after, inter-helix bridges form, thus allowing the formation of a water-bridge network into the collagen matrix. The water-bridge network allows the proton conduction into the film through the Grotthuss mechanism. Therefore, the ionic conductivity of the pure water-impregnated membrane is due to the transfer of protons which are intrinsically present in the polymeric matrix. After impregnation with H_2_SO_4_ solutions, a strong improvement of proton conductivity is visible, which is imputable to the increase in the number of membrane protons that can be transferred.

As a result, the obtained ionic-conductivity values for the H_2_SO_4_-impregnated membranes are suitable for application as electrolytes in electrochemical devices. In particular, considering the ionic conductivity and the mechanical characterization, we evaluated that the cross-linked membrane impregnated with H_2_SO_4_ 1% (*v*/*v*) represents the best option. Regarding optical properties, the total transmittance and the haze of the wetted membrane (with H_2_SO_4_ 1% *v*/*v*) were measured in the visible range of the electromagnetic spectrum; the results have been reported in Figure 7B. An excellent global transmittance was measured, with a mean transmittance value higher than 90% in the range of 400–800 nm. Concerning haze, it is used to measure the milky or cloudy appearance that is due to the scattering of light in transparent material. The lower the haze, the higher the clarity of the film, a desirable feature for application in electrochromic devices, where high transparency/clarity is required in the bleached state. The mean haze value of the cross-linked membrane is about 15% in the range of 400–800 nm, which is suitable for the mentioned application.

Considering the promising characteristics of the cross-linked membrane, it was tested as an electrolyte in an electrochromic device. The cross-linked collagen membrane impregnated in H_2_SO_4_ 1% (*v*/*v*) was sandwiched between glass/ITO/PEDOT:PSS and glass/ITO/SnO_2_ (Figure 8A). In this structure, PEDOT:PSS acts as an electrochromic material while SnO_2_ is the ion storage layer; both materials were deposited by solution processes [4]. In Figure 8D, the electromagnetic spectra in the visible range of this device at different applied voltages have been reported. In the bleached state, the device shows good transparency with a mean transmittance higher than 70% in the 400–800 nm range. When a negative voltage is applied, a reduction of the transmittance is visible, and an absorption band, peaked at about 650 nm, appears [49]. Such absorption band is typical of PEDOT:PSS, and it is imputable to the intercalation of H^+^ into the PEDOT:PSS thin film. The maximum ΔT is about 35% at about 650 nm (at −3V), and such transmittance variation can also be appreciated in Figure 8B,C, where the device is shown in the bleached and colored state. The kinetics of the device was also evaluated by measuring the transmittance at 650 nm, applying a square-wave potential signal from −3V to +1V and back. The results are reported in Figure 8E. The device shows very fast coloration and bleaching kinetics with τc (coloration time) equal to 3 s and τb (bleaching time) equal to 2 s, which confirm the good performance of the selected electrochromic material but also the excellent ionic-conduction properties of the collagen membrane electrolyte.

## 3. Conclusions

Bio-based polymers represent a more sustainable and eco-friendly alternative to conventional polymers in various fields of applications. In this work, the possibility of employing collagen membranes as hydrogel-polymeric electrolytes in full-solid-state electrochemical devices has been investigated. Indeed, collagen can absorb large amounts of water, and thus it could be employed as a polymeric matrix that can retain large quantities of aqueous electrolyte and can be used as a self-standing electrolyte membrane. Uncrosslinked and cross-linked collagen membranes were obtained by air-drying, and a detailed FTIR characterization was performed to evaluate the effectiveness of the cross-linking and the preservation of typical triple-helical conformation of collagen after cross-linking.

The stability and the mechanical properties of the membranes after impregnation in water and aqueous electrolyte were tested; the resulting measures demonstrated that cross-linked membranes could absorb the aqueous electrolyte and preserve their integrity and mechanical characteristics. The ionic conductivity of the cross-linked membranes after impregnation with the aqueous electrolyte (1%, 2%, and 5% *v*/*v* H_2_SO_4_) was measured by EIS spectroscopy, showing a high ionic-conductivity value, close to 10^−4^ S/cm for all the tested concentrations.

Considering these results and the good optical characteristic of the collagen membranes in terms of transparency and low haze, we tested the H_2_SO_4_-impregnated membrane as an electrolyte in an electrochromic device. The glass/ITO/PEDOT:PSS/impregnated collagen/SnO_2_/ITO/glass device showed a maximum ΔT of about 35% at about 650 nm (at −3 V) and a very fast kinetic with τc equal to 3 s and τb equal to 2 s. The good performances of the fabricated electrochromic device demonstrated that collagen represents a good candidate for the future development of greener full-solid-state electrochemical devices.

## 4. Materials and Methods

### 4.1. Materials

Insoluble fibrillar type I collagen from the equine tendon in dry flake form was provided by Typeone Biomaterials Srl (Lecce, Italy). Commercial ITO-covered glass was provided by VisionTek Systems Ltd. PEDOT:PSS (Clevios FET) was purchased from Heraeus (Hanau, Germany). UV curing resin (NOA65) was provided by Norland. Distilled water was obtained from Millipore Milli–U10 water purification facility from Merck KGaA (Darmstadt, Germany). All other chemicals used were of analytical grade and purchased by Sigma-Aldrich (St. Louis, MO, USA).

### 4.2. Collagen Membranes and Collagen Electrolyte Preparation

Collagen-thin membranes were developed by air drying following a previously optimized protocol [31,50]. In brief, a collagen suspension of 10 mg/mL in 0.5 M acetic acid was prepared, degassed under a vacuum, and cast in Petri dishes. Then, air drying was performed in a laminar flow hood for 72 h at room temperature [50]. Following air drying, collagen membranes (uncrosslinked) were peeled from the Petri dishes and exposed to dry heat at 121 °C for 72 h, under vacuum (*p* < 100 mTorr) (cross-linked) [20,31].

### 4.3. Electrochromic Device Fabrication

An electrochromic device with the following structure was prepared: glass/ITO/PEDOT:PSS/Collagen electrolyte/SnO_2_/ITO/glass. SnO_2_ was deposited starting from a colloidal solution in H_2_O (15%) by spin coating it at 2000 rpm for 1 min; after that, an annealing treatment at 250 °C for 30 min was performed. PEDOT:PSS was employed as an electrochromic material starting from a commercial solution; it was spin-coated onto ITO-coated glass substrate at 2000 rpm for 1 min and annealed at 140 °C for 20 min. An oxygen plasma treatment (50 W, 30 sccm, 2 min) has been performed on ITO-coated glass before PEDOT:PSS and SnO_2_ deposition.

The cross-linked collagen membrane was soaked in H_2_SO_4_ 1% (*v*/*v*) overnight, it was recovered from the solution, the excess liquid was dried with absorbent paper, and it was placed on the surface of the pre-patterned PEDOT:PSS (in the center of the glass/ITO substrate). NOA65 UV curing resin was deposited on the edges of the substrate, and subsequently, the glass/ITO/SnO_2_ substrate was placed on top of it. The device was then exposed to UV light (5 W UV lamp) to allow the glue to cure and seal the device.

### 4.4. Collagen Membrane and Collagen Electrolyte Characterization

FT-IR was performed using FTIR-6300 from Jasco GmbH (Pfungstadt, Germany) on 1 × 1 cm collagen membranes. Absorption spectra were recorded in the range 4000–400 cm^−1^ at a resolution of 4 cm^−1^ and smoothed according to the Savitsky–Golay method [31,40]. Three samples for each sample type were scanned, and each spectrum was collected as an average of 64 scans. Amide I was deconvoluted in the five Gaussian sub-bands of the β-sheet, random coil, α-helix, β-turn, and β-antiparallel contributes according to the second derivative analysis. The contribution of each sub-band was calculated as a percentage of the total amide I area. Lastly, amide A was deconvoluted in the four Gaussian sub-bands, according to the second derivative analysis, corresponding to the different OH bonding lengths. The contribution of each sub-band was calculated as a percentage of the total amide A area. The ratio of the methylene (-CH_2_) and methyl (-CH_3_) groups was assessed, as well as the ratio between amide III and the contribute at 1450 cm^−1^. Origin software from OriginLab Corporation (Northampton, MA, USA) was used for data analysis.

The swelling tests were performed by weighing the samples before and after overnight soaking in water and H_2_SO_4_ solutions (1%, 2%, and 5% *v*/*v*) at room temperature. The thickness and width of dry and wet specimens were measured using a Dino-Lite digital microscope (AnMo Electronics Corporation, New Taipei City, Taiwan). The experiment was performed in triplicate for each sample type. All data were expressed as mean ± the standard deviation. The statistical significance of experimental data was determined using the t-Student test. Differences were considered significant at *p* < 0.05.

The mechanical properties of the collagen substrates in water and H_2_SO_4_ solutions (1%, 2%, and 5% *v*/*v*) were evaluated using a ZwickiLine universal testing machine (Zwick/Roell, Ulm, Germany) equipped with a loading cell of 1 kN by tensile test. Swelled samples of 5 × 20 mm were clamped and tested under displacement control till failure with a preload of 0.1 N and a load speed of 0.1 mm/s [31,40]. The Young’s modulus (E), the stress at break (σ_max_), and the strain at break (εr) of samples were taken into account [20,31]. The experiment was performed in triplicate for each sample type. All data were expressed as mean ± the standard deviation. Differences were considered significant at *p* < 0.05 using the *t*-Student test.

Electrochemical impedance spectroscopy was used to measure ionic conductivity by using an Autolab PGSTAT 302 N Potentiostat/Galvanostat (Metrohm). The measurements were carried out under steady-state conditions (0 V) by applying an AC voltage of 0.01 V over a 100 kHz to 0.1 Hz frequency range. The membranes were sandwiched between two parallel platinum foils (area = 1 cm^2^), and the ionic conductivity (σ) was calculated by using the following formula:σ = s/(R · A)(1)
where s is the electrolyte thickness, A is the active area of the electrode/membrane/electrode system, and R is the real part of the impedance extracted from the intercept of the low-frequency signal in the Nyquist plot with the x-axis. The experiment was performed in triplicate for each sample type.

Total optical transmittance and haze spectra of the wetted cross-linked collagen membrane were measured by a Varian 5000 spectrophotometer.

### 4.5. Electrochromic Device Characterization

Optical transmittance was measured by a Varian 5000 spectrophotometer in a wavelength range between 300 nm and 800 nm by applying different voltages at the device with a Keithley Sourcemeter 2420. Kinetic measurements were performed by changing the potential between 1 V and −3 V using an Autolab PGSTAT302 N (Metrohm AG, The Netherlands) potentiostat.

## Figures and Tables

**Figure 1 gels-09-00310-f001:**
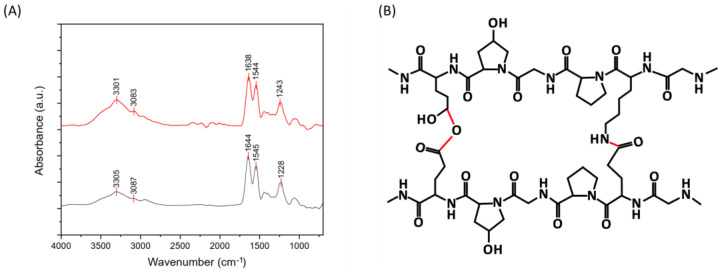
(**A**) FT-IR spectra of uncrosslinked (down) and cross-linked (up) collagen membranes. (**B**) Chemical structure of collagen after cross-linking.

**Figure 2 gels-09-00310-f002:**
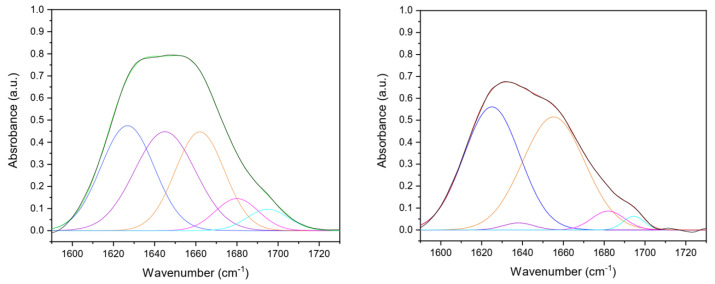
Gaussian deconvolution of UN (**left**) and DHT (**right**) collagen film amide I (1600–1700 cm^−1^) into four component bands, that were β-sheet (blue line, 1610–1642 cm^−1^), random coil (purple line, 1642–1650 cm^−1^), α-helix (orange line, 1650–1660 cm^−1^), β-turn (pink line, 1660–1680 cm^−1^), and β-antiparallel (teal line, 1680–1700 cm^–1^) components were detected. The fitted curve is reported with a green (UN) or red (DHT) line, closely overlapping the experimental curve (black line).

**Figure 3 gels-09-00310-f003:**
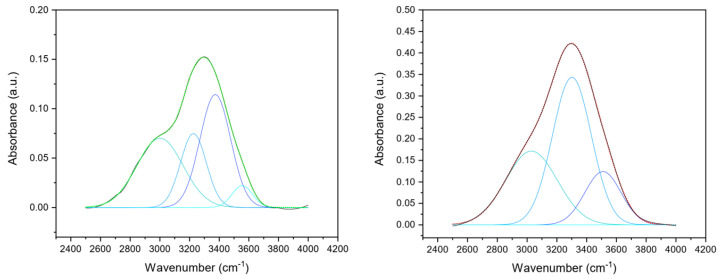
Gaussian deconvolution of FTIR spectrum of UN (**left**) and DHT (**right**) collagen film in the ν (OH) region in four components, corresponding to the H-bond distance of 0.25 nm (octane line, at about 3110 cm^−1^), 0.28 nm (sky blue line, at about 3250 cm^−1^), 0.29 nm (blue line, at about 3460 cm^−1^) and 0.31 nm (teal line, at about 3670 cm^−1^). The fitted curve is shown as the green (UN) or red (DHT) line, closely overlapping the experimental curve (black line).

**Figure 4 gels-09-00310-f004:**
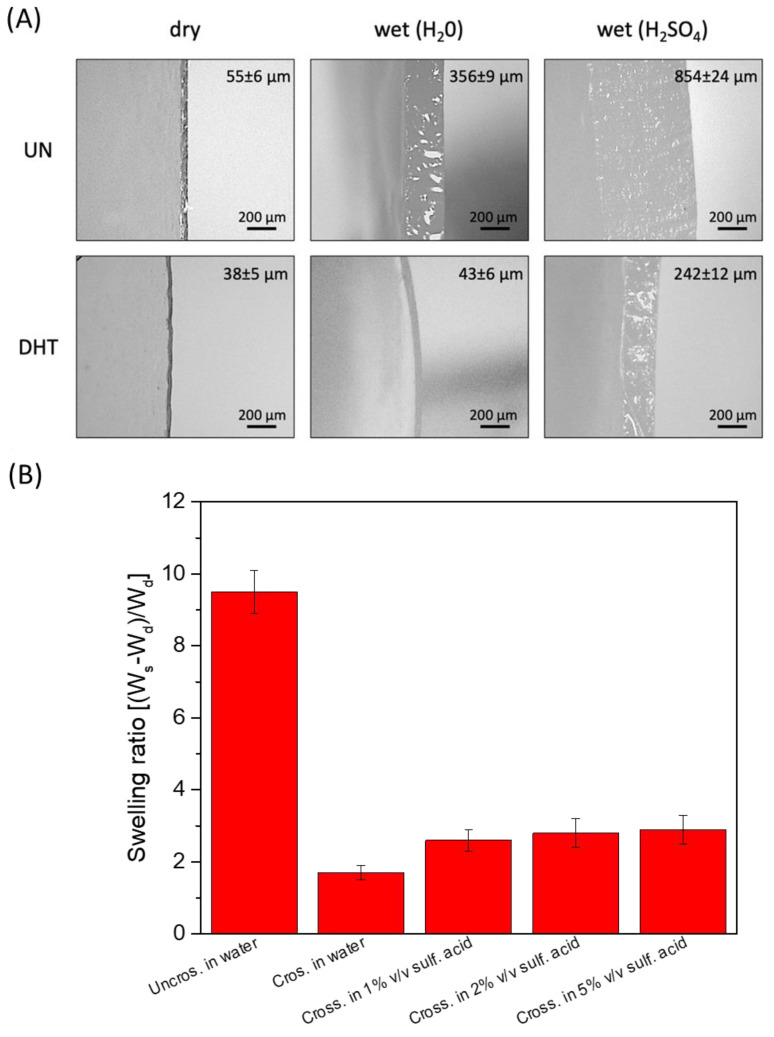
(**A**) Representative snapshot of uncrosslinked and cross-linked collagen matrices thickness in a dry state and in the presence of water and of sulfuric acid 1% (*v*/*v*). (**B**) Swelling degree of uncrosslinked and cross-linked collagen matrices thickness in a dry state and in the presence of water and of 1–5% (*v*/*v*) H_2_SO_4_.

**Figure 5 gels-09-00310-f005:**
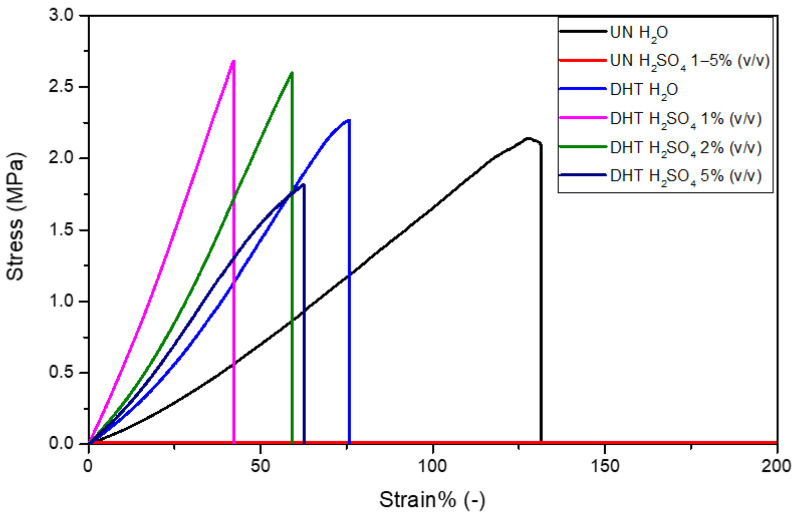
Representative stress-strain curves of uncrosslinked and cross-linked collagen matrices in the presence of water and sulfuric acid 1–5% (*v*/*v*).

**Figure 6 gels-09-00310-f006:**
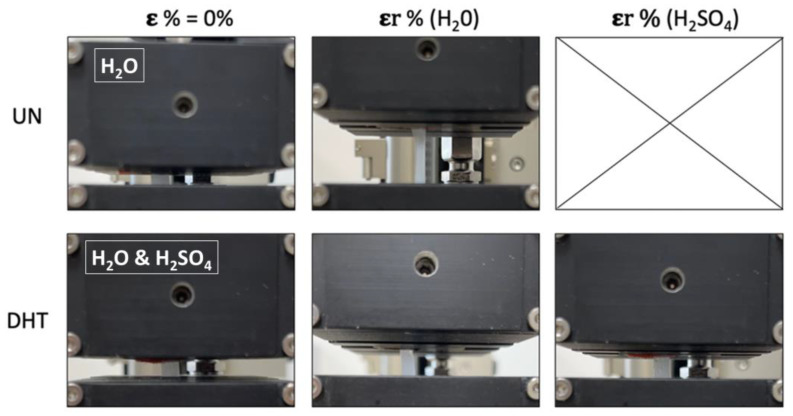
A representative snapshot of uncrosslinked and cross-linked collagen matrices at ε% = 0% and at their maximum deformation before the break in the presence of water and aqueous electrolytes (the snapshot of the UN membrane at ε% = 0% is referred to the sample impregnated with H_2_O since with H_2_SO_4_ it was not possible to perform measurements; the snapshot of the DHT membrane at ε% = 0% is referred to sample impregnated with both H_2_O and H_2_SO_4_ since no significant differences were visible).

**Figure 7 gels-09-00310-f007:**
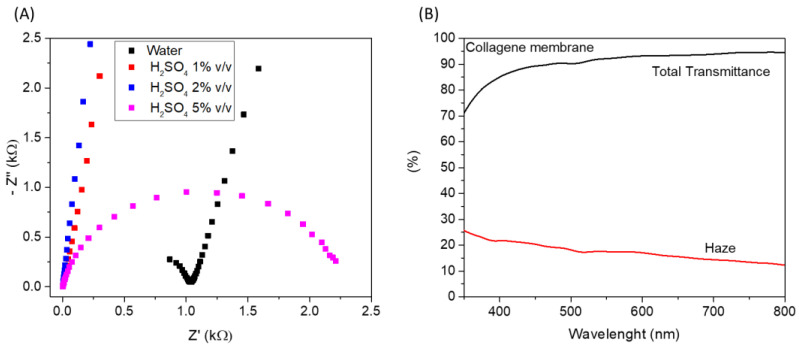
(**A**) Nyquist plots of electrochemical impedance spectra of the different membranes measured at 0 V; (**B**) Total transmittance and haze of cross-linked collagen membrane impregnated with H_2_SO_4_ 1% (*v*/*v*).

**Figure 8 gels-09-00310-f008:**
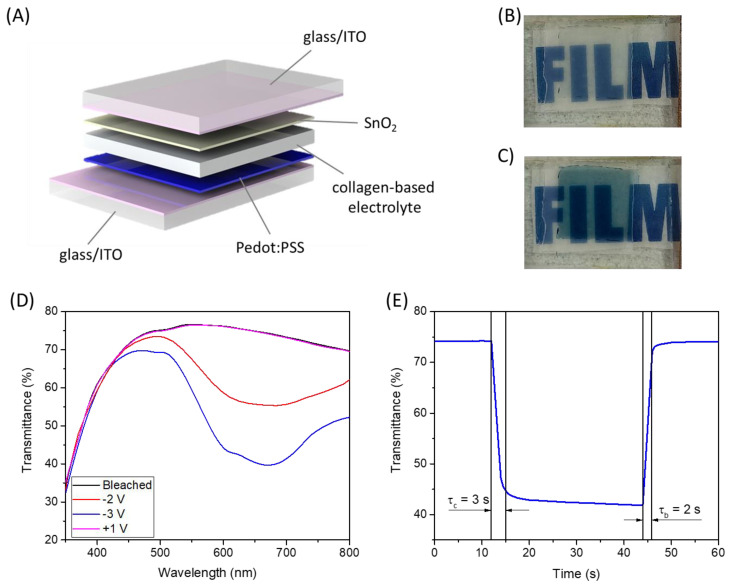
(**A**) Schematic representation of the fabricated electrochromic device; (**B**) picture of the device in the bleached state; (**C**) picture of the device in the colored state; (**D**) electromagnetic spectra of the electrochromic device in the visible range at different applied voltages; (**E**) transmittance (650 nm) vs. time spectrum by changing the potential between 1 V and −3 V.

**Table 1 gels-09-00310-t001:** Secondary structure peaks and percentage analysis of uncrosslinked and cross-linked collagen matrices in the 1600–1700 cm^−1^ spectral range.

Sample	β-Sheets 1610–1642 cm^−1^	Random Coil 1642–1650 cm^−1^	α-Helices 1650–1660 cm^−1^	β-Turn 1660–1680 cm^−1^	β-Antiparallel 1680–1700 cm^−1^
UN film	1626 (29.8%)	1645 (32.1%)	1660 (26.0%)	1679 (7.4%)	1695 (4.7%)
DHT film	1625 (45.8%)	1638 (1.6%)	1655 (46.6%)	1681.9 (4.1%)	1694.4 (1.9%)

**Table 2 gels-09-00310-t002:** Hydrogen bond distances of uncrosslinked and cross-linked collagen matrices in the 2400–4000 cm^−1^ spectral range.

Stretching Frequency (1/cm)	ν1	ν2	ν3	ν4
H-bond distances (nm)	0.31	0.29	0.28	0.25
UN film	3673 (4.4%)	3490 (18.6%)	3250 (50.6%)	3116 (26.4%)
DHT film	3671 (0.0%)	3482 (16.8%)	3249 (50.0%)	3135 (33.2%)

**Table 3 gels-09-00310-t003:** Mechanical properties of uncrosslinked and cross-linked collagen matrices in the presence of water and of aqueous electrolytes in terms of Young’s modulus (E), maximum stress (σ_max_), and strain at break (εr). Reported values represent mean ± SD, where n = 3.

Sample Type	E (MPa)	𝛔max (MPa)	𝛆r% (-)
Uncosslinked collagen membrane (H_2_O)	1.2 ± 0.3	1.8 ± 0.6	131 ± 7
Uncosslinked collagen membrane (1% H_2_SO_4_)	-	-	-
Uncosslinked collagen membrane (2% H_2_SO_4_)	-	-	-
Uncosslinked collagen membrane (5% H_2_SO_4_)	-	-	-
Cross-linked collagen membrane (H_2_O)	2.7 ± 0.5	2.3 ± 0.3	67 ± 7
Cross-linked collagen membrane (1% H_2_SO_4_)	4.7 ± 0.6	2.3 ± 0.4	40 ± 8
Cross-linked collagen membrane (2% H_2_SO_4_)	2.8 ± 0.5	2.7 ± 1.2	65 ± 15
Cross-linked collagen membrane (5% H_2_SO_4_)	2.5 ± 0.1	1.5 ± 0.4	55 ± 11

## Data Availability

The data presented in this study are available on request from the corresponding author.

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
