# Peer review of "Collagen Membrane as Water-Based Gel Electrolyte for Electrochromic Devices"

_gels, 2023, doi:10.3390/gels9040310_

Round 1
Reviewer 1 Report
Authors fabricated collagen membrane as water-based gel electrolyte for electrochromic devices. The as-fabricated device based on the H2SO4 impregnated membrane shows good performances. There are some errors in the article that need to be corrected.
1. It is recommended to visually draw the differences in the structure or composition of the UN and DHT membranes using schematic diagrams as figure 1 in order to highlight the innovative points.
2. The SEM images in Figure 4 should appear in the text as Figure 1.
3. The horizontal and vertical coordinates in the Raman diagram should be changed to “cm-1” and “a.u.”.
4. It is recommended to increase the font size in the figures (especially in Figure 4b) to increase readability.
5. It is recommended to consider citing the following literature: 10.1016/j.jpowsour.2022.231684
6. In figure 6, the image of UN in Li2SO4 should not appear.
7. In figure 7a, the range of values for the real and imaginary axes should be equal.
8. In figure 8, the figure number C is missing.
Author Response
Reviewer #1
Authors fabricated collagen membrane as water-based gel electrolyte for electrochromic devices. The as-fabricated device based on the H2SO4 impregnated membrane shows good performances. There are some errors in the article that need to be corrected.
- It is recommended to visually draw the differences in the structure or composition of the UN and DHT membranes using schematic diagrams as figure 1 in order to highlight the innovative points.
We thank the Reviewer for the comment. Accordingly, an explicative representation was added in Figure 1 to show the new bonds formed in the crosslinked collagen and highlight UN and DHT structural differences.
- The SEM images in Figure 4 should appear in the text as Figure 1.
We thank the Reviewer for such comment. We cited membrane thickness at the beginning of the “Results and Discussions” section and so the mentioned figure should appear close to such paragraph. Nevertheless, we prefer to maintain Figure 4A and 4B in the same panel since they show two different characteristics of the same swelling process. Therefore, we decided to delete the reference to the thickness at the beginning of the “Results and Discussions” section and to discuss this aspect later.
- The horizontal and vertical coordinates in the Raman diagram should be changed to “cm-1” and “a.u.”.
We modified the labels in Figure 1-2-3 as suggested by the Reviewer and the new figures are reported in the revised manuscript.
- It is recommended to increase the font size in the figures (especially in Figure 4b) to increase readability.
We thank the Reviewer for this suggestion. We improved the font size of the graphs and the new figures are reported in the revised manuscript.
- It is recommended to consider citing the following literature: 10.1016/j.jpowsour.2022.231684
We thank the Reviewer for the advice. We added such literature work in the introduction part of the revised manuscript.
- In figure 6, the image of UN in Li2SO4should not appear.
We thank the Reviewer for this comment and we apologize if the figure was not fully understandable. We tried to better clarify what is represented in Figure 6. As mentioned in the text, we weren’t able to perform tensile tests with UN membrane impregnated with H2SO4. Therefore, the snapshot of the UN membrane at ε % = 0% is referred only to the sample impregnated with H2O, while the snapshot of the DHT membrane at ε % = 0% is referred to the sample impregnated with both H2O and H2SO4, since no significant differences were visible. We added such details in the figure caption and we added labels to the figure in the revised manuscript.
- In figure 7a, the range of values for the real and imaginary axes should be equal.
We modified the range values in Figure 7a and the new figure is reported in the revised manuscript.
- In figure 8, the figure number C is missing.
We added the C letter in Figure 8 and the new figure is reported in the revised manuscript
Reviewer 2 Report
The authors have presented the investigation on the fabrication and the characterization of uncrosslinked and physically crosslinked collagen membranes. Furthermore, physico-chemical properties of both the membranes such as the membranes stability in water and aqueous electrolyte and the mechanical properties were evaluated. Currently, the growing concern on the climate change has bring the attention back to utilization of biomaterials in various fields to reduce the dependency on chemical entities. In this regard, the applications and investigations of biopolymers in different fields including electrochemical devices is a welcome step. The manuscript is well presented with sufficient data and the materials were thoroughly characterized. However, there are certain shortcomings which must be addressed before the acceptance of the article. Such as,
1. The authors should clearly explain the purpose of studying uncrosslinked and physically crosslinked collagen membranes in the introduction (difference and advantages of each)
2. Too many introductory statements in the beginning of the abstract given an impression that the article is not original research but a review, these statements should be reduced
3. Sentence in the line 22 to 24 should be rephrased
4. Figure 4A would be well suited in the beginning of the results and discussion section as a separate figure.
5. Figure 4B should be replaced with another clear image, the quality is not good.
6. Section 4.1 “gently provided” is a wrong statement
7. The manuscript is well written, however, at several places various typographical and grammatical mistakes have been found. Few of them are mentioned above, there many more like, thus the manuscript should be thoroughly checked for these errors
Author Response
Reviewer #2
The authors have presented the investigation on the fabrication and the characterization of uncrosslinked and physically crosslinked collagen membranes. Furthermore, physico-chemical properties of both the membranes such as the membranes stability in water and aqueous electrolyte and the mechanical properties were evaluated. Currently, the growing concern on the climate change has bring the attention back to utilization of biomaterials in various fields to reduce the dependency on chemical entities. In this regard, the applications and investigations of biopolymers in different fields including electrochemical devices is a welcome step. The manuscript is well presented with sufficient data and the materials were thoroughly characterized. However, there are certain shortcomings which must be addressed before the acceptance of the article. Such as,
- The authors should clearly explain the purpose of studying uncrosslinked and physically crosslinked collagen membranes in the introduction (difference and advantages of each)
We thank the Reviewer for the constructive criticism. Collagen was selected as alternative, eco-friendlier material for the development of electrochemical devices. However, as-produced collagen membranes are characterized by a very low degradation resistance in acid environments. To this, among crosslinking strategies, a physical crosslinking method was chosen and applied in order to enhance collagen substrates properties without reducing their eco-friendly character and their biocompatibility profile. In particular, a heat-mediated crosslinking method was selected to induce the formation of amide and ester bonds via condensation reactions.
We added the following paragraph in the introduction part of the revised manuscript:
“Because of as produced uncrosslinked collagen membranes’ low degradation resistance in acid environments, a physical crosslinking method was chosen and applied among crosslinking strategies to enhance collagen substrates properties without reducing their eco-friendly character and their biocompatibility profile. In particular, a heat-mediated crosslinking method was selected to induce the formation of amide and ester bonds via condensation reactions [29].”
- Too many introductory statements in the beginning of the abstract given an impression that the article is not original research but a review, these statements should be reduced
We thank the Reviewer for this suggestion. We modified the abstract and reduced the introductory statements in the revised manuscript.
- Sentence in the line 22 to 24 should be rephrased
We thank the Reviewer for this advice. We rephrased the sentence in the revised manuscript.
- Figure 4A would be well suited in the beginning of the results and discussion section as a separate figure.
We thank the Reviewer for the suggestion. We believe that figure 4A and 4B should be evaluated in the same panel in the manuscript since they show two different information (thickness variation and weight variation) related to the swelling behavior of the material. To improve the readability of the manuscript, we have eliminated the thickness information in the initial part of the “Results and Discussion” and this information is debated later in the revised manuscript.
- Figure 4B should be replaced with another clear image, the quality is not good.
We modified Figure 4 in the revised manuscript.
- Section 4.1 “gently provided” is a wrong statement
We thank the Reviewer for the comment. Accordingly, we removed the wrong statement.
- The manuscript is well written, however, at several places various typographical and grammatical mistakes have been found. Few of them are mentioned above, there many more like, thus the manuscript should be thoroughly checked for these errors.
We thank the Reviewer for the suggestion. We double-checked the manuscript and corrected typographical and grammatical mistakes in the revised version.
Round 2
Reviewer 1 Report
Authors have addressed the comments well